# Effectiveness of blended learning in pharmacy education: A systematic review and meta-analysis

**Athira Balakrishnan**[1], **Sandra Puthean**[2], **Gautam Satheesh**[2], **Unnikrishnan M. K.**[2,3], **Muhammed Rashid**[1], **Sreedharan Nair**[1], **Girish Thunga**[1]*

**1** Department of Pharmacy Practice, Manipal College of Pharmaceutical Sciences, Manipal Academy of Higher Education, Manipal, Udupi, Karnataka, India, **2** Department of Pharmacy Practice, National College of Pharmacy, Kozhikode, Kerala, India, **3** NGSM institute of Pharmaceutical Sciences, NITTE University, Manglore, Karnataka

* girish.thunga@manipal.edu

**Data Availability Statement:** All relevant data are within the paper and its Supporting Information files.

**Funding:** No fund received.

## Abstract

### Background & objective

Though blended learning (BL), is widely adopted in higher education, evaluating effectiveness of BL is difficult because the components of BL can be extremely heterogeneous. Purpose of this study was to evaluate the effectiveness of BL in improving knowledge and skill in pharmacy education.

### Methods

PubMed/MEDLINE, Scopus and the Cochrane Library were searched to identify published literature. The retrieved studies from databases were screened for its title and abstracts followed by the full-text in accordance with the pre-defined inclusion and exclusion criteria. Methodological quality was appraised by modified Ottawa scale. Random effect model used for statistical modelling.

### Key findings

A total of 26 studies were included for systematic review. Out of which 20 studies with 4525 participants for meta-analysis which employed traditional teaching in control group. Results showed a statistically significant positive effect size on knowledge (standardized mean difference [SMD]: 1.35, 95% confidence interval [CI]: 0.91 to 1.78, p<0.00001) and skill (SMD: 0.68; 95% CI: 0.19 to 1.16; p = 0.006) using a random effect model. Subgroup analysis of cohort studies showed, studies from developed countries had a larger effect size (SMD: 1.54, 95% CI: 1.01 to 2.06), than studies from developing countries(SMD: 0.44, 95% CI: 0.23 to 0.65, studies with MCQ pattern as outcome assessment had larger effect size (SMD: 2.81, 95% CI: 1.76 to 3.85) than non-MCQs (SMD 0.53, 95% CI 0.33 to 0.74), and BL with case studies (SMD 2.72, 95% CI 1.86–3.59) showed better effect size than non-case-based studies (SMD: 0.22, CI: 0.02 to 0.41).

**Competing interests:** The authors have declared that no competing interests exist.

## Conclusion

BL is associated with better academic performance and achievement than didactic teaching in pharmacy education.

## Introduction

Evaluating the effectiveness of blended learning (BL), a thoughtful combination of both online and face-to-face instructions, is difficult because the components of BL can be extremely heterogeneous [1, 2]. For instance previous systematic reviews / meta-analyses on BL have included multiple techniques such as virtual face-to-face interaction, simulations, online instruction, e-mails, computer laboratories, mapping and scaffolding tools, computer clusters, interactive presentations, handwriting capture, class room web sites, and virtual apparatuses [3]. Also, there is no standardized proportion in which BL combines online with face-to-face instructions [4].

Flipped learning 'and 'hybrid learning' are often used interchangeably with BL. In flipped learning, the learner is first exposed to online content, which will be reinforced during face-to-face sessions [5]. Hybrid learning, a combination of face-to-face instruction with computer mediated instruction, is most often used in United States [6]. In all forms of BL, the learner enjoys a certain degree of autonomy in deciding the pace of learning. However, previous reported systematic reviews on BL have not taken the keyword "flipped" in their search strategy [7, 8].

Increased research has been published on BL in medical education over last decades. For instance, Quian Liu et al's systematic review and meta-analysis reported that BL has consistent positive effects in comparison with no intervention for knowledge acquisition in the health professions [7]. In another systematic review, McCutcheon et al reported a deficit of evidence on implementation of BL in undergraduate nursing education [9]. Most of the published systematic review and meta-analyses in medical education were focused on medical students or nursing students or other healthcare professionals [8–10]. There is only one meta-analysis that evaluated the effectiveness of flipped learning in pharmacy education, with a major limitation namely, lack of prospective randomized control trials (RCT) and restrictions to the domain of flipped contexts [11]. Accordingly, we designed our objective to assess the effectiveness of BL which employed a combination of online and face-to-face instruction in blended, hybrid and flipped contexts in pharmacy education. We have considered BL as a combination of online and face-to-face instruction, excluding other computer mediated forms like virtual labs, gamifications, simulations to limit heterogeneity and included all possible synonyms of blended, hybrid, flipped learning and pharmacy education.

## Materials and methods

This study followed Preferred Reporting Items for Systematic Reviews and Meta-Analyses (PRISMA) Guidelines (PRISMA Checklist attached in S1 Appendix).

### Eligibility criteria

We employed PICOS (population, intervention, comparison, outcome, and study design) framework for the inclusion of studies. Studies were considered eligible, if they: (1) were conducted among pharmacy students, (2) used a BL intervention in the experimental group, (3)

used traditional lecture based learning as control for two arm studies and pre-test score for single arm studies (4) reported knowledge score/ objective structured clinical examination (OSCE) score as outcome (5) were two-group controlled studies (randomised/non-randomised)/ single group pre-test-post- test studies.

We excluded studies which did not explicitly state components of BL i.e. face-to-face learning and computer assisted learning. Computer assisted learning can be any form of technologies like online learning, e-learning, video podcasts, or the application of university learner management system for posting lectures. We excluded studies which employed "virtual face-to-face" interactions (as practiced by universities with satellite campuses). Studies which did not report a quantitative outcome of knowledge (comparison of students who completed and did not complete online module, number of correct answers between the groups, comparison of pass percentage), studies which evaluated only online component of BL, and surveys were also excluded. Multi-year studies without differentiating between study term years were excluded. Reviews, short communication, conference proceedings, editorials, meeting abstracts and non-English studies were also excluded.

## Data sources and literature search

A literature search employing PubMed, Scopus and Cochrane Library, was performed using a comprehensive search strategy since the inception of each database up to mid-December 2020. We employed all the MesH terms and key words for "BL" (Blended learning, blended course, blended program, hybrid learning, hybrid Course, Hybrid Program, Hybrid training, Flipped learning, Flipped Course, Flipped Program, Computer-aided learning, Computer-assisted learning, Integrated learning, Distributed learning, Distributed education Integrated instruction, Computer-aided instruction, Computer-assisted instruction) and "Pharmacy Student" which was obtained from the databases and previous studies. We employed the asterisk (*) as a wildcard character in keyword searches. We also searched for additional reference materials by consulting the cross references listed in the included publications, in addition to Google and Google Scholar (Details in S2 Appendix).

## Study selection and data extraction

The retrieved studies from databases were screened for its title and abstracts followed by the full-text in accordance with the pre-defined inclusion and exclusion criteria (List of excluded studies provided in S3 Appendix). We compiled and collated data in a comprehensive data extraction form containing characteristics such as, author and year of publication, population, duration and subject covered, nature of BL, sample size, and outcomes. The above data extraction form was perfected by trial and error, by piloting on 3 articles. Three independent reviewers were involved in study selection and data extraction to limit the bias and any disagreements were resolved through consensus or by discussion with another member of research team.

## Quality assessment

Modified Newcastle Ottawa scale (Newcastle Ottawa scale-education) was used to appraise methodological quality of included studies [12–14]. This tool assessed the following criteria: 1) representativeness of intervention group (1 point) 2) selection of comparison group (1point) 3) comparability of comparison group (2 point) 4) study retention (1 point) 5) blinding of assessment (1 point), totalling a maximum of 6 points. Two independent reviewers were involved to appraise the methodological quality to limit the bias and any disagreements were resolved through consensus or by discussion with another member of research team.

## Data synthesis

The evidence were synthesized narratively and presented in tabular form. We employed meta-analysis whenever possible. We omitted studies from data pooling whenever data did not meet the requirements of meta-analysis, such as, participant number, mean and standard deviation [SD]. All comparisons were based on scores of consecutive years. If more than one topic was delivered by BL in same study with separate scores for each, we considered them as separate studies. RevMan 5.3 was used to conduct the meta-analysis [15]. The data were used as mean with SD and outcomes were presented as standardised mean difference (SMD) along with 95% confidence interval (CI). Studies that did not report a SD, the corresponding SD from the p-values and standard errors were generated as per Cochrane guideline [16]. Heterogeneity was assessed by $I^2$ statistics and random effect model used for statistical modelling. Subgroup analysis were performed to find out potential source of heterogeneity based on factors like studies with case studies and without case studies, studies which reported outcome as a measure of multiple choice questions(MCQs) or non MCQs, and studies from developed and developing countries. Sensitivity analysis were performed to ensure the robustness of findings.

## Publication bias

We employed a funnel plot for visual inspection of publication bias, which was assessed for statistical significance by Egger's and Begg's test [16].

## Results

A total of 2539 records were retrieved first, of which 2448 underwent initial screening. Next, 2383 studies were omitted, yielding 65 full-text studies, of which 26 studies were included for systematic review, and 20 for meta-analysis (See Fig 1 for details of study selection).

### Characteristics of studies included for systematic review

Of the 26 studies included, only two employed single arm pre-test-post-test design [17, 18]. The remaining 24 studies were controlled studies [19–42] out of which 19 used examination scores of previous year [19–34, 36–41] and one used examination score of subsequent year as control [35]. There were 3 randomised trials [14, 19, 31] out of which one was cluster randomised [24]. Another study divided learning materials into didactic and BL in same population [28]. 18 studies originated from USA and 8 studies from other countries [17, 20, 21, 23–26, 28, 33] (See Table 1 for characteristics of included studies).

### Outcome measured

Only 3 studies [18, 19, 39] reported outcome as skills(patient centred interpersonal communication skills, students' performance on pharmaceutical calculation, and critical care therapeutics) while 21 studies reported only knowledge score [17, 20–29, 31–36, 38, 40–42]. Two reported both knowledge and skills as outcomes [30, 37]. Outcomes were measured variably as mean examination percentage (n = 16) or mean examination score (n = 6) or objective structured clinical examination (OSCE) (n = 2). Two studies reported both examination percentage and OSCE score.

### BL approaches

Two studies employed face-to-face session followed by online activities [17, 34] while all other studies employed face-to-face session after watching online content. Only one study reported time spent and workload associated with BL [37].

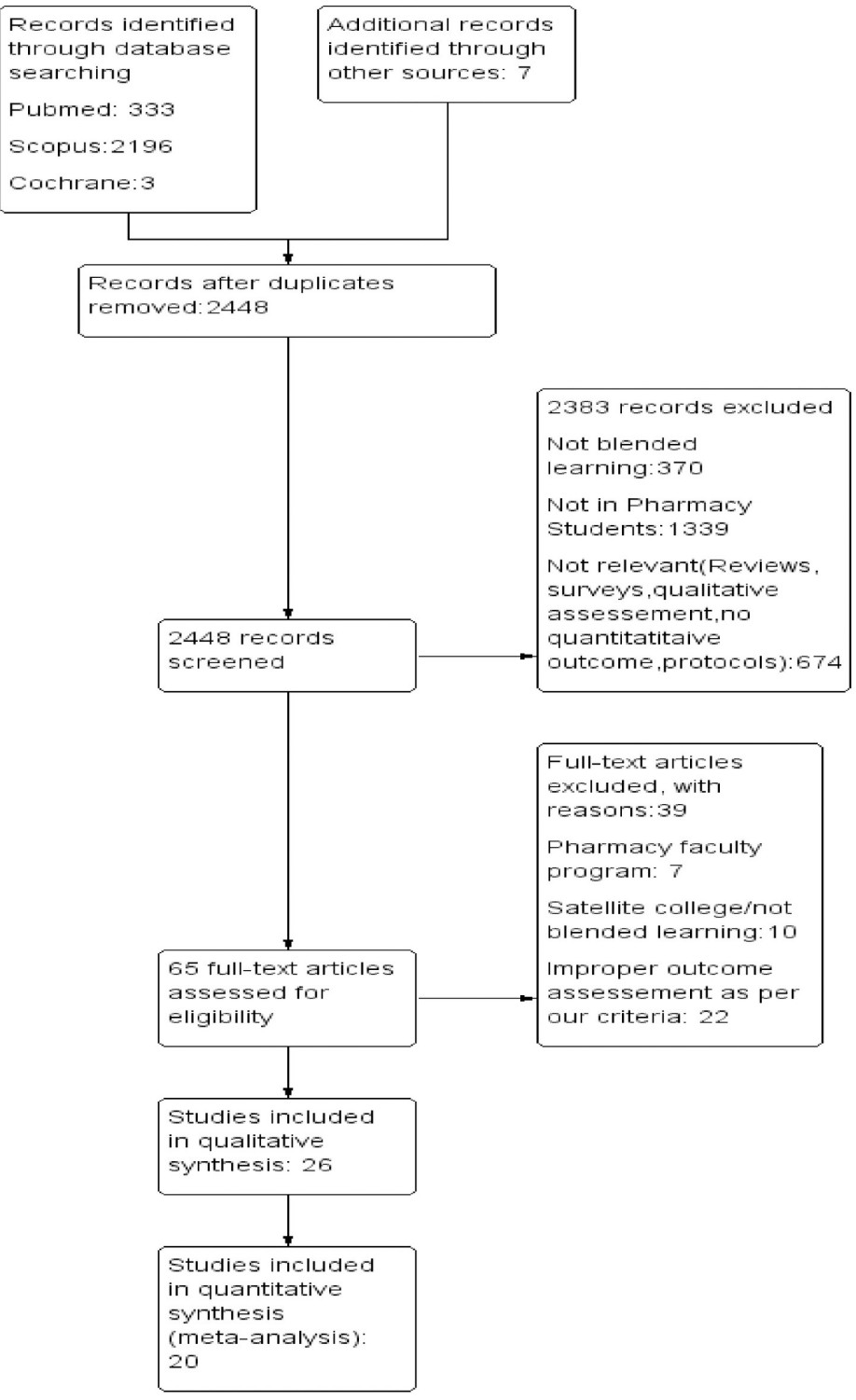

**Fig 1. PRISMA flow diagram.**

**Table 1. Characteristics of included studies.**

| Author | Country | Population | Topic(duration) | Intrv. details | Study design | Sample size Intrv. | Sample size Control | Post-intrv. academic outcome | Other activities | Mean Result Score Intrv. | Mean Result Score Control | Major outcome |
|---|---|---|---|---|---|---|---|---|---|---|---|---|
| Wilson et al [40] (2019) | USA | 2nd year pharmaco-therapy students | Selected self-care pharmaco-therapy (NC) | Online (Vimeo) + class activities | Cohort (compared with previous year students' score) | N/A | N/A | Exam performance (65% course grade)-Assessment questions | TBL | 83.5% | 83.3% | No statistically significant differences in student outcomes |
| Newsom et al [34] (2019) | USA | 1st year students enrolled in spring 2015–17. (Control: spring '14) | Pharmacokinetics (NC) | Traditional class and video podcast | Cohort Intrv.: 2015–17 Didactic teaching: 2014 | 2015: 153 2016: 152 2017: 153 | 2014: 175 | Final exam score: questions based on Bloom's taxonomy. | Case based practice problems | 2015: 85.8(7.7) 2016: 85.1(9.2) 2017: 78(12) | 2014: 77.6 (13.3) | Final exam scores were significantly higher in spring '15 and '16 compared to '14 (p<0.001). 2017 scores were similar to that of 2014. |
| GoH et al [23] (2019) | Malaysia | 2nd year Dosage Form II course | 3 credit course. Dosage form II (NC). | Pre-recorded video + F2F sessions | Cohort: Two group comparison ('16 & '17 batch) | 63 | 74 | Final Exam score. Subjective (5 MCQ + 2 essay) | Online games | 49.93 | 41.24 | Final exam performance significantly higher in the flipped classroom group |
| He et al [24] (2019) | China | Junior year pharmacy under-graduates | Pharma-ceutical marketing (4 months) | Online + class | Cluster randomization | 81 | 56 | Final score (subjective– short answer, essays, MCQ) | Case discussion Group activity | 88.21 ± 5.95 | 80.05 ± 5.59 | Compared with LBL methods, implementing the FC model improved student performance. |
| Kouti et al [28] (2018) | Iran | Pharmacy Students (2015–16 batch) | Non-prescription drugs (1 semester) | Electronic based + lecture based | Propective comparative study– 3-group study (f2f, Electronic, BL) | 57 | - | Final exam score (not clear) | Case studies | E-learning group: 16.17 ± 0.33; Lecture group: 13.75 ± 0.16; BL: 16.39 ± 0.19 | - | BL method and an e-learning approach can positively influence students' knowledge towards non-prescription drugs |
| Kangwantas et al [26] (2017) | Thailand | 2nd year pharmacy students | Fundamental nutrition (1 year) | Videos (moodle platform) +class activities | Cohort compared with previous year score. | 29 | 21 | Pre and post-test within the group; Post-test: main exam scores. (subjective– MCQ + short answer) | Case discussion | Pre-test: (7.45 ±1.89 and Post-test: (8.17±1.44) not statistically different (p = 0.08). | Flipped class scored higher (7.24±1.24 vs. 6.19±1.76) (p = 0.028) | Student performance as measured by final scores of the module was better than those for the same module taught with a traditional lecture in previous year |
| Koo et al [27] (2016) | USA | 2nd year PharmD students | Pharmaco-therapy (1 year) | Online + F2F discussion | Cohort-comparison (2011 & 2012) | 89 | 89 | Exam score: Objective, MCQ | Case study discussion | 88.2% (7.3%) | 83.4% (7.9%) | The redesigned course improved student test performance and perceptions of learning experience |
| Giuliano et al [22] (2016) | USA | 1st year Pharmacy Students | Drug literature evaluation (1 year) | Youtube lecture +Class session. | Cohort: 2-group study (2013 & 2014) | 94 | 99 | Final exam score: subjective-application, analysis & Bloom's taxonomy evaluation | Group activities | 86.1% | 75.6% | The flipped model is an excellent fit for drug literature content and courses that want to incorporate more active learning |

(Continued)

**Table 1.** (Continued)

| Author | Country | Population | Topic(duration) | Intrv. details | Study design | Sample size Intrv. | Sample size Control | Post-intrv. academic outcome | Other activities | Mean Result Score Intrv. | Mean Result Score Control | Major outcome |
|---|---|---|---|---|---|---|---|---|---|---|---|---|
| Edginton et al [21] (2013) | Canada | 2nd year pharmacy students | Bio-chemistry (1 year) | online + classroom | Cohort: 2011 vs. 2010 | 116 | 109 | Final grade: subjective-MCQ + calculation + long answer | Group discussion; Case studies | 78.8 + 11.7 | 61.8 + 17.8 | The student driven BL model correlated positively with increased interest and perceived and actual learning gains. P<0.00001. |
| Pierce [36] (2012) | USA | Pharmacy students | Renal pharmaco-therapy (8 weeks) | Video podcast + classroom | Pre-test and post-test-within the group (2012) / Only post-test between groups (2012 & 2011) | 71 | N/A | Objective-MCQ | Case discussion | Pre-test (33.5 ± 11.6 and post-test (79.2 ± 10.6): within the group; Between groups: 81.6 ±4.4 | 77.7 ± 4.7 | Implementing a flipped classroom to teach renal pharmacotherapy resulted in improved student performance and favourable student perceptions |
| McLaughlin et al [31] (2015) | USA | PharmD students | Neuro-logic pharmaco-therapy (NC). | e-learning + class | Randomized (same class) | 57 | 59 | Final exam score: 9 final exam questions- not clear whether questions are MCQ/subjective. | Case studies | 80.12 + 13.57 | 74.76 + 15.12 | Interactive online preparatory tool improves student learning in neurologic pharmacotherapy. |
| Wong et al [41] (2014) | USA | 1st year pharmacy students | Cardiac arrhythmia (3 classes) | Pre-recorded video + class | Cohort: compared with previous year (2012 & 2011) | 101 | 103 | Final exam score: 5–6 MCQ on cardiac arrhythmias. | Case based exercises | Basic science: 88.3±1.9; Pharmacology: 89.6±2; Therapeutics: 89.2±1.4 | 84.1 ± 1.9; 56.8 ± 2.2; 73.7 ± 2.1 | Use of the flipped teaching in a 3-class pilot on cardiac arrhythmias improved exam scores for pharmacology and therapeutics classes. |
| Anderson et al [19] (2017) | USA | 1st year pharmacy students | Pharma-ceutical calculations (6 weeks) | recorded lecture + video | Randomized | 38 | 32 | Final exam- Skill: OSCE Score at 6 weeks | Case studies | 71.3 (14.7)% | 61.8 (17.7)% | Average OSCE performance was be higher in flipped model than lecture model |
| Cotta et al [20] (2016) | Georgia | 1st year pharmacy students | Pharma-ceutical calculation (10 weeks) | Pre-recorded video + class | Cohort: 2012 vs. 2011 | 151 | 165 | Final exam part 2 score- objective graded quizzes | - | 88.3 (9.5) | 84.1 (11.3) | Flipped classroom can improve student performance and satisfaction in pharmaceutical calculations (P<0.001) |
| Lancaster [29] (2011) | USA | 2nd PharmD | OTC medicines (15 weeks) | Pre-recorded video + class | Cohort: 2008 vs. 2009 | 97 | 97 | Final Exam scores- objective-9 quizzes | Clinical based cases with Group discussion; Puzzles, Think pair share activities. | 84.09 | 65.15 | Students performed significantly higher on quizzes and examinations when using this hybrid teaching model. |

(Continued)

**Table 1.** (Continued)

| Author | Country | Population | Topic(duration) | Intrv. details | Study design | Sample size Intrv. | Sample size Control | Post-intrv. academic outcome | Other activities | Mean Result Score Intrv. | Mean Result Score Control | Major outcome |
|---|---|---|---|---|---|---|---|---|---|---|---|---|
| Stewart [38] (2013) | USA | 3rd year pharmaco-therapy | Pharmaco-therapy (NC) | Podcast + active learning | Cohort: 2009 vs. 2010 | 71 | 65 | Final exam score: 20 MCQ | Group discussion | 72.9+1.5 (12.63) | 77.15+1.2 (9.6) | The class averages on the final exams were significantly higher for 2009 batch compared to 2010 batch (P: 0.019). |
| Lockman et al [30] (2017) | USA | 1st year pharmaco-logy & therapeutics course | Pain management module (NC) | E learning + in class lecture | Cohort: 2015 vs. 2016 | 162 | 156 | OSCE: skill | Cases & mini cases, quiz games, Mind-mapping debates | MCQ: 82.30% (SD 10.25) OSCE: 79.34(9) | 77.23% (SD 12.43) OSCE: 67.01 (9.6) | Student performance improved significantly after flipping the content of pain management module. |
| | | | | | | | | Knowledge-MCQ | | | | |
| Nazar et al [33] (2018) | Qatar | Stage 2 Pharmacy under-graduate students | Pharmacy law (NC) | Online class + In-class activities. | Cohort study: compared with previous year | 69 (2016–17) | 63 (2015–16) | Final summative examination score | Group discussion | 82.2 (6.3)% | 84.2 (6.8)% | Examination performance appeared to be unaffected by the change in teaching style |
| | | | | | | | 37 (2014–15) | | | | 83.0 (7.6)% | |
| Hughes et al [25] (2016) | USA | P1 pharmacy students | Drug information (5 weeks) | Narrated video + F2F lab session | Cohort study: compared with previous years (2012 Vs. 2013) | 127 | 121 | Final exam score: Objective-40 MCQ | - | 88.99% | 84.87% | Mean final exam scores significantly increased (p < 0.05) |
| Gloudeman et al [42] (2017) | USA | 1st year pharmacy students | Pharmaceutical calculation (6 week) | Online video + classroom session | Cohort study: compared with previous years (2015 Vs. 2014) | 102 | 104 | Final exam score: 13 pharma-ceutical calculation questions | - | 80.5 ± 15.8% | 77.8 ± 16.8% | The mean exam scores of the intrv. were not significantly different than the control (p = 0.253) |
| Czepula et al [17] (2017) | Brazil | Under-graduate bachelor's degree in pharmacy | Pharma-ceutical care(NC) | F2F + distance learning | Quasi-experimental, prospective. Two groups. Both groups received BL of pharma-ceutical care 1 & 2. | Pharma-ceutical care 1: 82 Pharmaceutical care 2: 51 | | Pre- and post-test: 30 MCQ | - | Module 1: Mean scores increased from 4.8 to 6.3 (p<0.05) Module 2: Mean scores increased from 4.1 to 5.5 (p<0.05) | | Positive results were observed regarding the students' performance in the two disciplines |
| Prescott et al [37] (2016) | USA | 1st and 2nd year pharmacy students | Two course: PA1 and PA2(NC) | Online videos + class | Cohort study: Comparison of traditional and BL of PA1 & PA2 (2014–15 vs, 2013–14) | PA1: 130 PA2: 131 | PA1: 126 PA2: 122 | Final examination score: 20 short answer questions | Group discussion (TBL) Case based learning | Knowledge: PA1: 80.5 (9.6) PA2: 80.6 (14.3) P<0.001 Skill: PA1: 93.1 (7.6) PA2: 83.5 (12.5) P<0.001 | Know-ledge: PA1:73(12); PA2: 74.5(12.1) Skill: PA1: 89.1 (13.8) PA2: 81.5 (12.6) | BL was associated with improved academic performance and was received by students. |

*(Continued)*

**Table 1.** (Continued)

| Author | Country | Population | Topic(duration) | Intrv. details | Study design | Sample size Intrv. | Sample size Control | Post-intrv. academic outcome | Other activities | Mean Result Score Intrv. | Mean Result Score Control | Major outcome |
|---|---|---|---|---|---|---|---|---|---|---|---|---|
| Wanat et al [39] (2016) | USA | 3rd year pharmacy students | Critical care 2hr credit course(4 weeks) | Video recorded lecture + in-class activities | Cohort study: Compared with previous year score. (2013, 2014 Vs. 2012, 2011) | 51 | 54 | Overall exam performance: subjective: online quiz + skills: examining patients | Group learning | 87.7% (3.7) | 82.6% (6.3) | Exam scores of students in BL group is significantly higher than control |
| Phillips et al [35] (2016) | USA | 1st and 2nd year PharmD students | EBM & Therapeutics(6 month) | Online video prior to class room | Cohort study: Knowledge compared with previous year scores -two group comparison. | EBM: 201 Therapeutics: 199 | N/A | Final Exam score/-not clear whether questions are MCQ/ subjective. | | EBM:83% Therapeutics: 97 | EBM:85% Therapeutics: 98% | Use of the BLE did not seem to have an impact on long-term knowledge in this study |
| Hess et al [18] (2016) | USA | 2nd year pharmacy students | Patient centred communication skills (One semester,) | Online modules + small group discussion | Single group study | 57 | - | Pre-test and post-test OSCE | Group discussion | Significant Increase in scores from pre-test to post-test.7 domains of pre and posttest scores provided. | | Patient-centred interprofessional communication skills improved significantly with BL |
| McLaughlin et al [32] (2014) | USA | 1st year pharmaceutics students | Pharmaceutics course(NC) | Flipped classroom (iLAMs + F2F) | Cohort study: Traditional vs. Flipped 2012 vs. 2011 | 162 | 153 | Final exam grade. Subjective-quizes +examination scores | - | Final score; 165.48 ± 13.34 | 160.06 ± 14.65 | Higher final exam grades in flipped classroom |

Intrv.: Intervention; BL.: Blended learning, EBM: Evidence-based medicine; TBL: Team Based Learning; F2F: Face-to-Face, N/A-not Available, NC-not clear MCQ: Multiple choice questions, OSCE: Objective structured clinical examination, PA1: Patient assessment 1 course, PA2: Patient assessment 2 course, iLAMS.: integrated learning accelerator module.

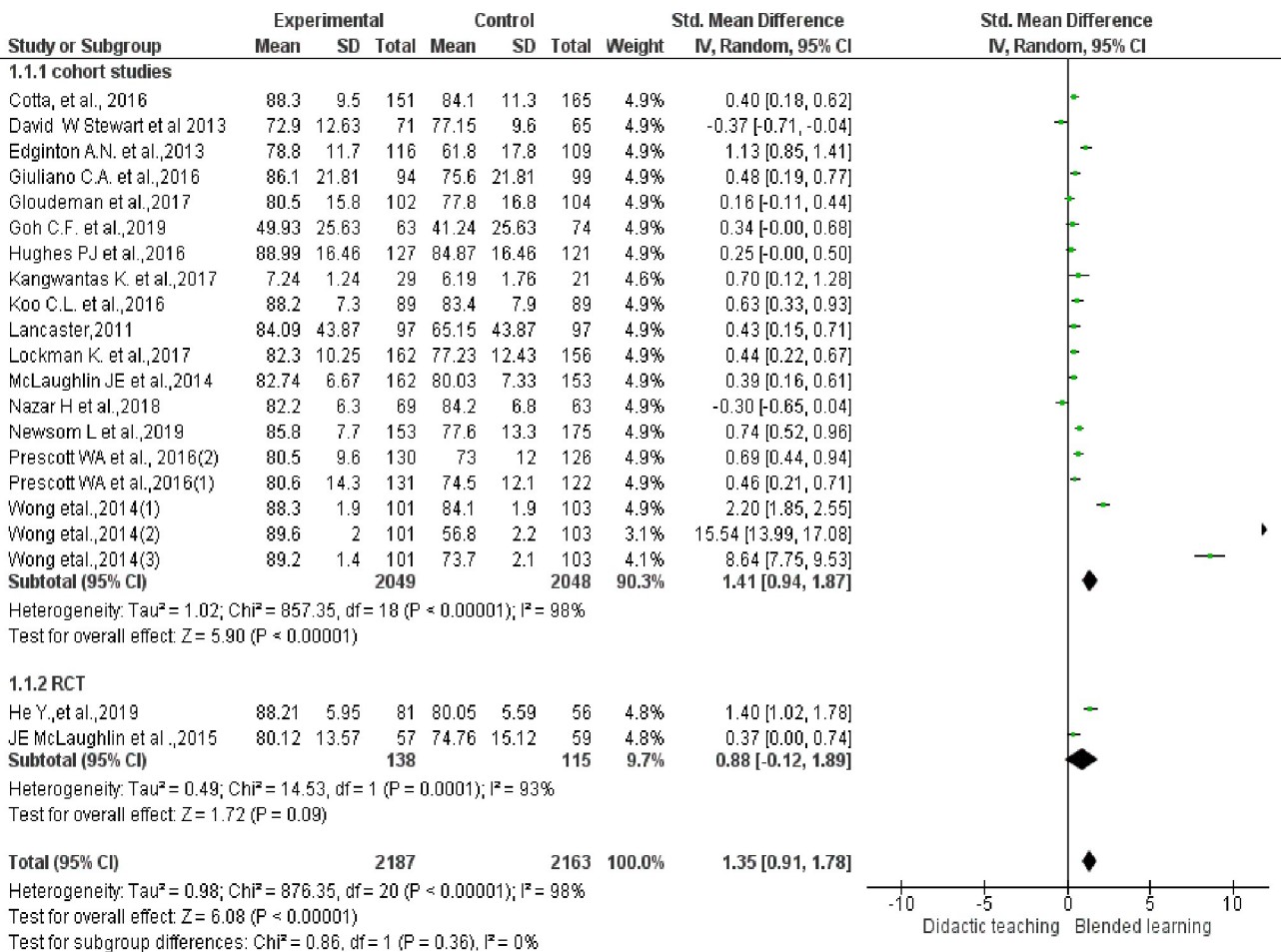

**Fig 2. Efficacy of BL vs. traditional teaching in improving knowledge.** If more than one topic was delivered by BL in same study (Prescott, Wong) with separate scores for each, we considered them as separate studies (Prescott 1&2, Wong 1, 2&3).

## Quality assessment of included studies

As per modified Ottawa scale requirements, we ascertained that intervention groups in all the included studies were representatives of target population. Out of 26 studies, 19 used previous year students' score as control, one used subsequent year score as control and 3 studies were randomized. Two studies used analysis of covariance(ANCOVA) for controlling covariates in final analysis [23, 38] and one used linear regression [22]. In five studies there were no statistically significant differences in students demographics / pre-test (Grade Point Average) between groups by t-test [27, 30, 32, 34, 41]. However, modified Ottawa scale requires controlling for subject characteristics by statistical covariate analysis. Outcome assessment was blinded for 11 studies, as assessor cannot be influenced by group assessment (third party statistician) or assessments did not require human judgments (MCQs/ graded performance) [17, 19–20, 25, 27, 29–30, 36, 38, 40–41]. As all studies were part of curriculum in educational institution, there is no mention about drop outs. All studies obtained a score below 4 except one [19] (See S4 Appendix).

## Quantitative analysis

We included 20 studies with 4525 participants for meta-analysis that employed traditional teaching in the control group and had no missing data.

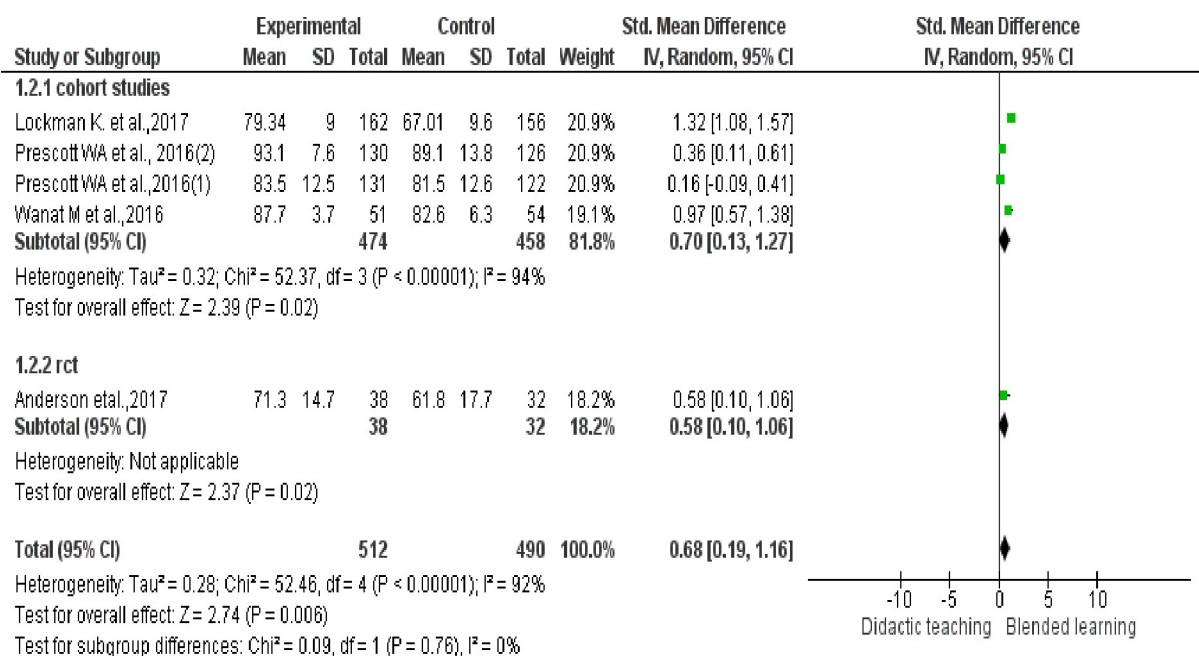

**Fig 3. Efficacy of BL vs. traditional teaching in improving skill.** If more than one topic was delivered by BL in same study (Prescott) with separate scores for each, we considered them as separate studies (Prescott 1&2).

## Efficacy of BL versus. Traditional teaching in improving knowledge

Pooled effect of 18 studies showed that knowledge improved significantly in BL, with large effect compared to didactic teaching ((SMD 1.35, 95% CI-0.91 to 1.78, p<0.00001). In the knowledge domain, randomised controlled studies had a lower pooled effect (SMD 0.88) than cohort studies (SMD 1.41). There was significant statistical heterogeneity among studies ($I^2$ = 98%, p<0.00001) with individual effect sizes ranging from −0.37 to 15.54 (See Fig 2).

## Efficacy of BL versus traditional teaching in improving skill

Pooled effect size (SMD 0.68, 95% CI: 0.19 to 1.16,Z = 2.74,p = 0.006) of 4 studies in improving skills, showed statistically significant moderate to large effect, compared with didactic teaching. Significant statistical heterogeneity was observed among studies ($I^2$ = 92%, p<0.00001) (See Fig 3).

## Subgroup analysis

Subgroup analysis of cohort studies, in the knowledge domain, demonstrated advantage for BL over traditional teaching, in developed countries (SMD 1.54, 95% CI 1.01–2.06) than developing countries (SMD 0.44, 95% CI 0.23–0.65). Studies which employed MCQ scores as outcome showed larger effect size (SMD 2.81, 95% CI 1.76–3.85) than non MCQs (SMD 0.53, 95% CI 0.33–0.74). Also, studies which employed case studies/case discussion favoured BL (SMD 2.72, 95% CI 1.86–3.59) than non-case based studies (SMD: 0.22, CI: 0.02 to 0.41). Subgroup analyses of studies improving skill were not performed, as all studies originated from United States of America and all employed case studies/case discussion. (See Table 2)

## Sensitivity analysis

A sensitivity analysis was performed in studies improving knowledge by removing two studies (Wong et al., [2, 3]) which are having lesser weight (3.1% and 4.1%, respectively), and higher

**Table 2. Subgroup analysis of cohort studies.**

| Study Characteristics: | Sample size | Test for heterogeneity | | | Test for effect | |
|---|---|---|---|---|---|---|
| | | I²(%) | Q statistics | P value | Pooled effect size(SMD(C1)) | P value |
| 1. Country | | | | | | |
| Developed | 3731 | 98 | 854.67 | P<0.00001 | 1.54(1.01,2.06) | P<0.00001 |
| Developing | 366 | 0 | 0.89 | P = 0.35 | 0.44(0.23,0.65) | P<0.0001 |
| Total | 4097 | 98 | 857.3 | P<0.00001 | 1.41(0.94,1.87) | P<0.00001 |
| 2 Outcome assessment | | | | | | |
| MCQ | 2002 | 99 | 796.46 | P<0.0001 | 2.81(1.76,3.85) | P<0.0001 |
| Non MCQ | 1635 | 76 | 29.47 | P<0.0001 | 0.53(0.33,0.74) | P<0.0001 |
| Not clear | 460 | 96 | 24.89 | P<0.00001 | 0.23(-0.80,1.25) | 0.66 |
| Total | 4097 | 98 | 857.35 | P<0.00001 | 1.41(0.94,1.87) | P<0.00001 |
| 3. Case studies | | | | | | |
| Present | 2364 | 99 | 736.66 | P<0.00001 | 2.72(1.86,3.59) | P<0.00001 |
| Absent | 1733 | 75 | 31.53 | P<0.0001 | 0.22(0.02,0.41) | 0.03 |
| Total | 4097 | 98 | 857.36 | P<0.00001 | 1.41(0.94,1.87) | P<0.00001 |

MCQ: Multiple choice questions; SMD: Standardised mean difference; CI: confidence interval.

outlier (MD: 15.54 and 8.64, respectively) which supported the main results (SMD: 0.55; 95% CI: 0.33 to 0.77). The result of sensitivity analysis is depicted in Fig 4.

## Publication bias

Visual inspection of funnel plot revealed an obvious asymmetry, demonstrating possible publication bias. This was confirmed by Egger's (P = 0.00006) and Begg's (P = 0.04) test (See Fig 5).

## Discussion

This systematic review and meta-analysis primarily attempted to evaluate the impact of BL approach on various outcomes in pharmacy education. We identified 26 studies relevant for systematic review, in which 18 demonstrated significant improvement in learning outcome, against controls. Two of them were single arm studies which also showed improved performance after intervention. 24 of the 26 studies included in this systematic review were controlled, among which majority (n = 19) employed examination scores of previous year(s) as the control. All studies employed first online review of contents followed by face-to-face discussion except two. Studies which employed face-to-face discussion followed by online activities also favoured BL [17, 34]. The face- to- face discussion part of BL in all included studies involved either reinforcing the concepts by tutor or using learning strategies such as case studies, case discussion or group activities.

In addition to the general scarcity of literature comparing BL and traditional methods, a major limitation of the previous meta-analysis by Gillette et al., was the lack of prospective RCTs [11]. Our meta-analysis included 20 of the studies included in the systematic review. Our review included 3 RCTs, all of which showed major improvements in either knowledge score or skill. We report a large pooled effect size for knowledge and a medium to large for skills. These findings were statistically significant with high heterogeneity in all analyses and are consistent with those reported by previous meta-analyses in medical education.

The majority of the studies reported knowledge score in terms of either mean examination percentage/score or OSCE, whereas 5 studies reported outcomes based on skill. Many of the studies included in this review also reports that BL has a major effect on improving teaching as

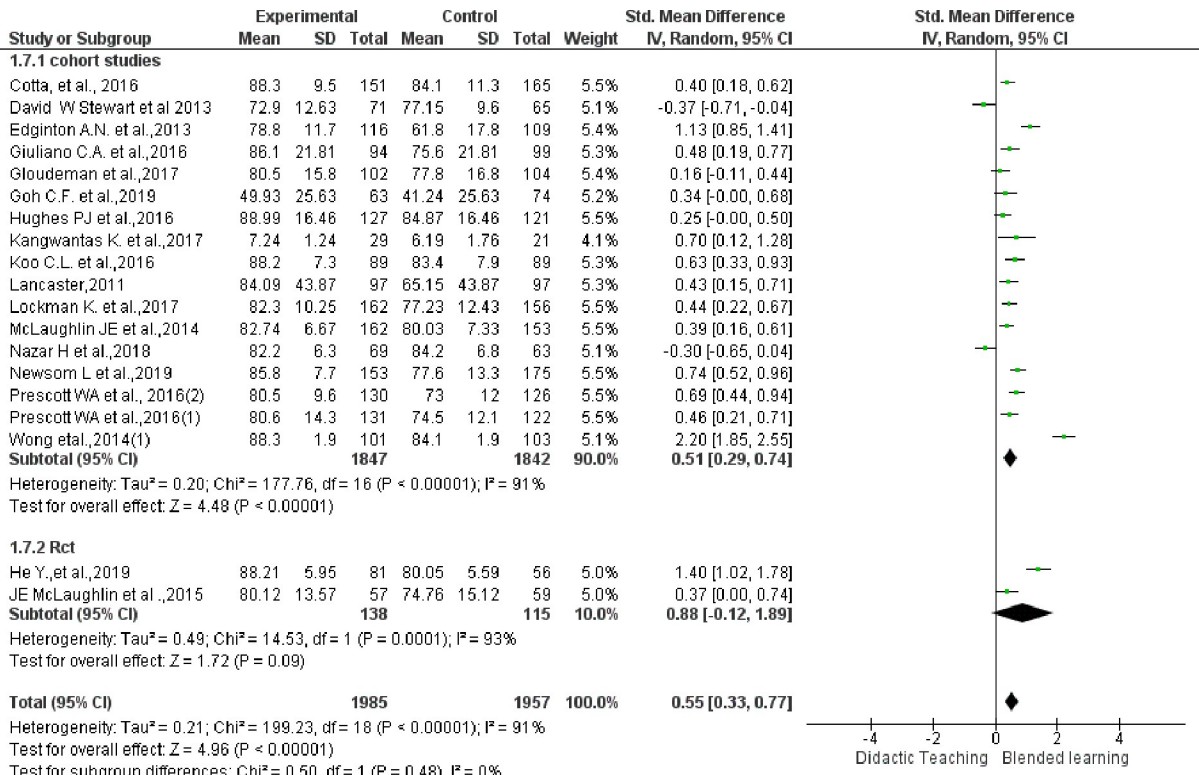

**Fig 4. Sensitivity analysis: If more than one topic was delivered by BL in same study (Prescott, Wong) with separate scores for each, we considered them as separate studies (Prescott 1&2).**

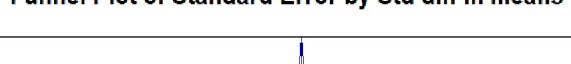
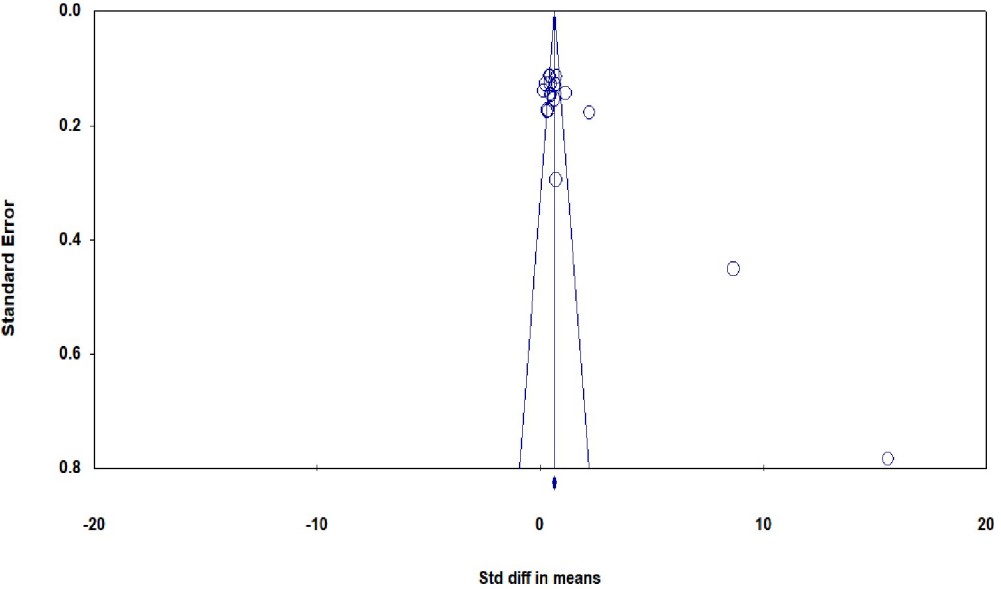

**Fig 5. Funnel plot of BL versus traditional teaching in improving knowledge.**

well as positive student perceptions about learning. As mentioned earlier, the rich variety of components can attribute to an enhanced learning experience as well as increased engagement and learning activities such as group assessment, assessment quizzes and peer discussions. Even the studies that did not report a significant difference in acquisition of knowledge–such as those by Phillips et al., and Gloudeman et al. showed that the perceptions of both students and faculty favoured BL [35, 42].

Another important finding is that BL modules which employed case studies/discussions or case-based scenarios reported better outcomes. A few studies also concluded that positive results obtained may not be attributed entirely to the suggest on that case studies need to be included in learning strategies [24, 37]. There is evidence to show that case studies simulate real world situations and enhance interactive student-centred learning, particularly in the health professions. Incorporating case studies in a real-world context is extensively useful in pharmacy education, as it enhances students' complex decision-making abilities.

Out of 26 studies, only 4 originated from developing countries, possibly because of poor online connectivity, lack of resources, fear of adopting unfamiliar technology, lack of skill development program to instructors, interruption in power supply and internet connections, affordability, low bandwidth and trust deficit [17, 20, 26, 28]. A single study that compares time budgets reported that BL techniques were completed ahead of allotted time [35]. BL approach appears to significantly improve the learning outcomes in pharmacy students and reason could be following,

i. **Relaxed/flexible scheduling**: BL allows students to view electronic materials at their own pace and time

ii. **Improved interaction**: BL makes classroom discussion more meaningful because of content familiarity.

iii. **Variety of components**: BL incorporates a rich variety of face-to-face and online components.

This study has a few limitations. First, the search was restricted to the publications in English language, which might have contributed to missing out eligible studies in non-English speaking countries. However, a comprehensive search in various databases would have covered the maximum quality publications. Second, our review also excludes conference proceeding and unpublished or grey literature. However, this may increase the credibility of our findings obtained from full length papers by avoiding the irrelevant or incomplete acquisition of the data. Third, there was high heterogeneity among the outcomes or measures of outcome, thereby restricting our choice exclusively to studies reporting quantitative outcomes. Fourth, the heterogeneous administration pattern of BL was an another challenge in this review, so we included those studies which used online teaching along with face-to-face approach, this made our result more robust and conclusive. Statistical heterogeneity was high in all analysis. However, this is in accordance with other meta-analysis in medical education [7, 8, 43]. Subgroup analyses did not find any source of heterogeneity. Despite the effective search strategy, one major limitation is the majority i.e. 18 of the 26 studies, were from the US, which could impact the global representativeness of the findings. Therefore, future research should address the impact of BL in diverse populations from other countries.

Publication bias was addressed by including the three major scientific databases (Pubmed, SCOPUS and Cochrane) during the literature search. This resulted in an increased number of papers which may have further increased the likelihood of selecting papers with negative results. In our review, 5 of the 26 studies reported that BL yields either equal or poorer outcomes than didactic teaching [33, 35, 38, 40, 42].

## Conclusion

BL is associated with better academic performance and achievement than didactic teaching in pharmacy education. The COVID-19 pandemic is radically reshaping the education sector to transform from conventional teaching to more online learning. In this scenario, it is critical to conduct more controlled empirical studies to evaluate the effectiveness of BL. Such research can inform education policies and guidelines to standardise blended learning.

## Supporting information

**S1 Appendix. PRISMA checklist.**
(DOCX)

**S2 Appendix. Search strategy in database.**
(DOCX)

**S3 Appendix. List of excluded studies.**
(DOCX)

**S4 Appendix. Quality assessment.**
(DOCX)

## Author Contributions

**Conceptualization:** Girish Thunga.

**Data curation:** Athira Balakrishnan, Girish Thunga.

**Formal analysis:** Athira Balakrishnan, Muhammed Rashid, Girish Thunga.

**Investigation:** Sandra Puthean, Gautam Satheesh.

**Methodology:** Athira Balakrishnan, Muhammed Rashid.

**Project administration:** Sandra Puthean.

**Software:** Athira Balakrishnan.

**Supervision:** Athira Balakrishnan, Unnikrishnan M. K., Sreedharan Nair, Girish Thunga.

**Validation:** Athira Balakrishnan, Sandra Puthean, Gautam Satheesh.

**Writing – original draft:** Athira Balakrishnan, Sreedharan Nair, Girish Thunga.

**Writing – review & editing:** Athira Balakrishnan, Unnikrishnan M. K., Sreedharan Nair, Girish Thunga.

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
