## [Decision Letter · Decision Letter 0]

6 Mar 2021

PONE-D-20-37558

Effectiveness of blended learning in pharmacy education: A systematic review and meta-analysis

PLOS ONE

Dear Dr. Thunga,

Thank you for submitting your manuscript to PLOS ONE. After careful consideration, we feel that it has merit but does not fully meet PLOS ONE’s publication criteria as it currently stands. Therefore, we invite you to submit a revised version of the manuscript that addresses the points raised during the review process.

We look forward to receiving your revised manuscript.

Kind regards,

Gwo-Jen Hwang

Academic Editor

PLOS ONE

Journal Requirements:

Reviewers' comments:

Reviewer's Responses to Questions

**Comments to the Author**

1. Is the manuscript technically sound, and do the data support the conclusions?

Reviewer #1: Partly

Reviewer #2: Yes

2. Has the statistical analysis been performed appropriately and rigorously? 

Reviewer #1: No

Reviewer #2: Yes

3. Have the authors made all data underlying the findings in their manuscript fully available?

Reviewer #1: Yes

Reviewer #2: No

4. Is the manuscript presented in an intelligible fashion and written in standard English?

Reviewer #1: Yes

Reviewer #2: No

5. Review Comments to the Author

Reviewer #1: First, thank you for submitting this review and meta-analysis to the journal. Please consider the following during the peer review process:

1. Abstract - You can remove "This review follows the PRISMA guideline." - PRISMA is a reporting standard, not a methodology. In the full text, you can certainly say your review adheres to the reporting standards recommended by the PRISMA statement.

2. Eligibility (Page 3) - Please clarify. You state in (3) "pre-test score for single arm studies" and (5) "were two groups" - so did you included single group papers or not? Based on results, you included 2 single arm studies, so (5) doesn't appear to be an accurate inclusion criteria.

3. Acronyms - Spell out all acronyms for first use. For all tables, please include the abbreviations and acronym definitions as a key.

4. Page 13 - "face-to-face" and "face to face" (be consistent)

5. Quality Assessment - A significant majority of studies used prior year scores as control with few studies attempting to control for confounding. Is it appropriate to pool the results of analyses?

6. Unequal weighted test scores - It appears by focusing on the mean difference, you equally weight 1 point on a 100 point exam vs. a 50 point exam, despite the point value being worth twice as much on the latter.

Reviewer #2: This study was a SR and MA that sought to assess the effectiveness of blended learning teaching strategies compared to traditional lectures. The study was well done. Methods were clear. Technical writing and grammar need to be revised. Attached is a file with grammatical revisions.

6. PLOS authors have the option to publish the peer review history of their article (what does this mean?). If published, this will include your full peer review and any attached files.

Reviewer #1: **Yes: **T. Joseph Mattingly II

Reviewer #2: **Yes: **Alexandra Perez

---

## [Author Response · Author response to Decision Letter 0]

20 Apr 2021

Reviewers comments:

Reviewer 1:

Abstract - You can remove "This review follows the PRISMA guideline." - PRISMA is a reporting standard, not a methodology. In the full text, you can certainly say your review adheres to the reporting standards recommended by the PRISMA statement-

Answer: 

Corrected as per reviewer

2. Eligibility (Page 3) - Please clarify. You state in (3) "pre-test score for single arm studies" and (5) "were two groups" - so did you included single group papers or not? Based on results, you included 2 single arm studies, so (5) doesn't appear to be an accurate inclusion criteria.

 Answer: 

Corrected.

 We have included interventional studies like single group pretest and post test studies , two group controlled studies(randomized& non randomized). If it is single grouped, we have taken only those studies with pretest and post test); If it is two group studies, didactic teaching as control. 5 points in eligibility criteria is based on PICOS framework.

3. Acronyms - Spell out all acronyms for first use. For all tables, please include the abbreviations and acronym definitions as a key. 

Corrected

4. Page 13 - "face-to-face" and "face to face" (be consistent)

Corrected

5. Quality Assessment - A significant majority of studies used prior year scores as control with few studies attempting to control for confounding. Is it appropriate to pool the results of analyses?

We have separately mentioned pooled effect size of studies with historical group as control and studies with RCT design. Based on previous literature, and Cochrane guideline, we understand nothing wrong to pool this way. We pooled both conditions separately to avoid or adjust this dissimilarity or heterogeneity in the form of a subgroup analysis. Also, we presented the result as an overall to know the effectiveness blended learning as whole. As per Cochrane recommendations, all eligible studies can be included in the meta-analysis, regardless of the risk of bias assessment. Indeed, since almost all studies have low score, Cochrane suggests to present an estimated intervention effect based on all available studies, together with a description of the risk of bias in individual domains.

Reference: (Higgins JPT, Green S. Cochrane Handbook for systematic reviews of interventions version 5.1.0.: the Cochrane collaboration 2011)

As per the Cochrane Guideline of systematic review (section 9.6), we need to perform a subgroup analysis by splitting the data according to some specific characters such as participant, intervention or publication characters etc., to explore the effect of that particular factor in the analysis. 

As its educational intervention study in pharmacy education, practically it is difficult to conduct prospective controlled trial in same batch of students in same academic institution. That’s why most of the studies adopted historical group as control. Only three studies are RCTs. Out of 26 studies, we included 20 studies with no missing data for quantitative analysis. Among them 8 studies controlled confounding variables (please see quality assessment of studies by modified Ottawa scale). However, modified Ottawa scale requires controlling for subject characteristics by statistical covariate analysis. We have assigned “0” score for studies that tried to compare the baseline characteristics (adjusting confounders) by any method other than statistical covariate analysis. In all those studies, there were no statistically significant differences in students demographics / pre-test (Grade Point Average) between groups by t-test also. 

6. Unequal weighted test scores - It appears by focusing on the mean difference, you equally weight 1 point on a 100 point exam vs. a 50 point exam, despite the point value being worth twice as much on the latter.

As per the Cochrane guideline, When studies have used different instruments to measure the same construct, a standardized means difference (SMD) may be used in meta-analysis for combining continuous data, hence we used SMD to combine our results. Reference link is provided.

https://handbook-5-1.cochrane.org/chapter_9/9_2_3_2_the_standardized_mean_difference.htm

Reviewer #2: 

This study was a SR and MA that sought to assess the effectiveness of blended learning teaching strategies compared to traditional lectures. The study was well done. Methods were clear. Technical writing and grammar need to be revised. Attached is a file with grammatical revisions

Grammatical errors corrected.

---

## [Decision Letter · Decision Letter 1]

17 May 2021

Effectiveness of blended learning in pharmacy education: A systematic review and meta-analysis

PONE-D-20-37558R1

Dear Dr. Thunga,

We’re pleased to inform you that your manuscript has been judged scientifically suitable for publication and will be formally accepted for publication once it meets all outstanding technical requirements.

Kind regards,

Gwo-Jen Hwang

Academic Editor

PLOS ONE

Additional Editor Comments (optional):

Reviewers' comments:

Reviewer's Responses to Questions

**Comments to the Author**

1. If the authors have adequately addressed your comments raised in a previous round of review and you feel that this manuscript is now acceptable for publication, you may indicate that here to bypass the “Comments to the Author” section, enter your conflict of interest statement in the “Confidential to Editor” section, and submit your "Accept" recommendation.

Reviewer #1: All comments have been addressed

Reviewer #2: All comments have been addressed

2. Is the manuscript technically sound, and do the data support the conclusions?

Reviewer #1: Yes

Reviewer #2: Yes

3. Has the statistical analysis been performed appropriately and rigorously? 

Reviewer #1: Yes

Reviewer #2: Yes

4. Have the authors made all data underlying the findings in their manuscript fully available?

Reviewer #1: Yes

Reviewer #2: Yes

5. Is the manuscript presented in an intelligible fashion and written in standard English?

Reviewer #1: Yes

Reviewer #2: Yes

6. Review Comments to the Author

Reviewer #1: No additional comments. Thanks for your submission to the Journal!

Reviewer #2: The authors still need to made some minor grammatical edits. After these corrections are made, it is ok to fully accept.

---

## [Editor Report · Acceptance letter]

8 Jun 2021

PONE-D-20-37558R1 

Effectiveness of blended learning in pharmacy education: A systematic review and meta-analysis 

Dear Dr. Thunga:

I'm pleased to inform you that your manuscript has been deemed suitable for publication in PLOS ONE. Congratulations! Your manuscript is now with our production department. 

Kind regards, 

on behalf of

Dr. Gwo-Jen Hwang 

Academic Editor

PLOS ONE